# Impact of the Mbarara University of Science and Technology residency training on increasing access to specialty care workforce

**Leevan Tibaijuka**[1]*, **Jonathan Kajjimu**[2], **Lorna Atimango**[2], **Asiphas Owaraganise**[3], **Adeline Adwoa Boatin**[4], **Musa Kayondo**[1], **Nixon Kamukama**[5], **Joseph Ngonzi**[1,2]

1 Faculty of Medicine, Department of Obstetrics and Gynecology, Mbarara University of Science and Technology, Mbarara, Uganda, 2 Faculty of Medicine, Mbarara University of Science and Technology, Mbarara, Uganda, 3 Infectious Diseases Research Collaboration, Kampala, Uganda, 4 Department of Obstetrics and Gynecology, Massachusetts General Hospital, Boston, Massachusetts, United States of America, 5 Mbarara University of Science and Technology, Mbarara, Uganda

* ltibaijuka@must.ac.ug

**Data Availability Statement:** The datasets used during this study are available and have been uploaded.

## Abstract

Graduate tracer studies provide an avenue for assessing the impact of residency training on the distribution and access to specialty care and exploring job and professional satisfaction of alumnus. This study examined how the Mbarara University of Science and Technology (MUST) clinical residency training program influenced the spatial distribution and career paths of specialists. We conducted a mixed methods study involving an online survey and 12 in-depth interviews (IDIs) from June to September 2022. The online survey was distributed to a convenient sample of clinical residency alumnus from MUST via email and Whatsapp groups. Alumnus were mapped across the countries of current work in QGIS (version 3.16.3) using GPS coordinates. Descriptive and thematic analyses were also conducted. Ninety-five alumni (34.3%) responded to the tracer survey. The majority were males (80%), aged 31–40 years (69%), and Ugandans (72%). Most graduated after 2018 (83%) as obstetricians/gynecologists (38%) and general surgeons (19%). There was uneven distribution of specialists across Uganda and the East-African community—with significant concentration in urban cities of Uganda at specialized hospitals and academic institutions. Residency training helped prepare and equip alumnus with competencies relevant to their current work tasks (48%) and other spheres of life (45%). All respondents were currently employed, with the majority engaged in clinical practice (82%) and had obtained their first employment within six months after graduation (76%). The qualitative interviews revealed the reported ease in finding jobs after the training and the relevance of the training in enhancing the alumnus' ability to impact those they serve in teaching, research, management, and clinical care. Graduates cited low payment, limited resources, and slow career advancement concerns. Residency training improves the graduates' professional/career growth and the quality of health care services. Strategic specialty training addressing imbalances in subspecialties and rural areas coverage could optimize access to specialist services.

**Funding:** This work was funded by the Mbarara University Faculty of Medicine Seed grant. LT is supported by the Mbarara University Research Capacity Initiative (MURCI) Program funded by the Fogarty International Centre of the National Institute of Health (NIH) under grant number D43TW011632. The funders had no role in study design, data collection and analysis, decision to publish, or preparation of the manuscript.

**Competing interests:** The authors declare that they have no competing interests.

# Background

Post graduate residency training in developing countries is an important component in narrowing the global health care workforce crisis [1,2]. Access to specialized care greatly improves quality and safety of health care services [3,4]. WHO also recommends availability of competent professionals to provide routine care and management of complications for patients as a standard for improving quality of care in health facilities [5,6]. This can only be achieved if doctors are well prepared for this task through residency training [7].

Uganda and the East African region have significant inequalities in specialized care, with generally few medical specialists, and uneven distribution of the different medical specialties [8,9]. The small number is partly due to the small number of medical specialists graduating from the training institutions and the exodus of the majority of specialists in search of better pay and improved working conditions [10–12]. This definitely impacts the health care delivery to the population and widens the global health workforce crisis [6,13].

Graduate tracer studies provide data to explain the link between study programs and the job market, by exploring the employment situation of graduates in addition to their professional and personal careers, this provides indicators for their professional performance [14–16]. To better understand the public health impact of the post graduate training program at MUST, we aimed to document the geospatial location/distribution of the different specialties of clinical postgraduate alumni after residency training at MUST, their integration into the job market, scope of practice and current job satisfaction.

# Materials and methods

## Ethics statement

The study was approved by the Research Ethics Committee of Mbarara University of Science and Technology under reference number MUST-2022-366. Individual informed consent was obtained from all study participants. The consent form ended with the following statement; *"Do you agree to participate in this survey? Yes/No (Check response below as stated). If you agree to participate, please continue to the survey. If not, you may exit at any time."* Study codes were used to ensure the anonymity of participants' data. Data generated from the study shall be used for research purposes only and shall be private and confidential at all times. All principles of data transfer and principles of protection of human research participants outlined in the Declaration of Helsinki were observed.

## Study setting and design

We conducted a descriptive cross-sectional and phenomenological study among the clinical postgraduate alumni of residency training programs of the Faculty of Medicine at Mbarara University of Science and Technology (MUST) from 20[th]June to 11[th] September 2022. The study employed mixed methods involving both quantitative and qualitative approaches—an online survey and in-depth interviews. MUST is a public university that was established in October 1989 and is located 250 kilometers from Kampala in Southwestern Uganda. MUST is affiliated to Mbarara Regional Referral Hospital (MRRH), a public health facility that serves about five million people, mainly from 10 catchment districts of southwestern Uganda and serves as a teaching hospital for MUST medical school. The clinical post-graduate programs carry out their clinical work activities at Mbarara Regional Referral Hospital. The clinical post-graduate program is run under the faculty of Medicine, which is the oldest and pioneer faculty of MUST and has grown over a period of 33 years. MUST under its Faculty of Medicine (FoM) is accredited to offer the following post-graduate programs; Obstetrics and Gynecology,

Internal Medicine, Pediatrics and Child health, General Surgery, ENT, Pathology, Psychiatry, Dermatology, Emergency Medicine (2017), Anesthesia and Critical Care, Ophthalmology and Radiology. The programs are delivered through blended clinical clerkships, didactic courses and self-directed learning to ensure acquisition of competency-based knowledge and clinical skills relevant to the respective work environments.

## Study population and eligibility criteria

This study targeted and included all clinical postgraduate alumni after residency training at MUST. We included residency alumni who consented to and responded to the survey.

## Sample size estimation and sampling

We enrolled a convenience sample of all clinical residency alumni from MUST. The sample population included all residency alumni on the current electronic mailing lists of MUST residency alumni obtained from the academic registrar's office and the Directorate of graduate studies (DRGT) from the time of start of the graduate training at MUST (2003) to 2017 (graduates of the year 2021). In-depth interviews (IDI) participants were conveniently sampled from survey participants who expressed willingness to be interviewed, following the researchers' judgment during the study period to ensure a mix of specialties and years of completion of the residency training.

## Data collection

A questionnaire was designed using Google Forms and shared the link with the residency alumni of MUST. The sample population included all residency alumni on the current electronic mailing lists of MUST residency alumni obtained from the academic registrar's office and the Directorate of graduate studies (DRGT) from the time of start of the graduate training at MUST (2003) to 2017 (graduates of the year 2021). The online tracer survey was distributed to all former clinical residency graduates from MUST via email and Whatsapp platforms of clinical post-graduate alumni weekly for a period of 1 month using a Google Forms link. We obtained the following information from the online surveys; background socio-demographic information: age, sex, marital status, year of completion of residency program; transition into the labor market/ employment related characteristics: Employment situation: current job, employer, duration of current job, income, location of employment, duration from completion of residency to obtaining first job, previous employment, how many jobs switched to ever since graduation from residency, career/job satisfaction, search for employment, impact of skills obtained to the community. A Likert scale was used to assess the data on the relevance of the residency training and satisfaction with their career and professional situation. Respondents filled the form online and responses were automatically captured in Google Forms. The responses were downloaded as a CSV file.

The survey's last item asked if participants would be willing to be contacted for an interview. In-depth interviews (IDI) participants were randomly identified from survey participants who expressed willingness to be interviewed. The randomly identified participants were emailed to arrange an interview date and time. Interviews were conducted over a secure zoom link. These semi-structured interviews aimed to deeply explore the experiences of the alumni. Each interview started by the interviewer introducing themselves and reading out the following statement: *I will ask you several questions about your experiences following your completion of your postgraduate residency training at MUST, your transition to employment, your career and professional satisfaction during your current employment, and your impact to the community you serve following your residency training. Example questions asked included* "How and

when did you find your current and first job?" "How easy or difficult was it to find you first job?" "How much did you need residency training to do the work you are currently doing?" "How much are you putting the skills learnt during the programme to use?" "How much do the skills learnt during the training programme impact the community you serve?" Interviews ended after clarifying all questions had been answered and participants indicated they had nothing else to share. The interviews were transcribed verbatim and reviewed for accuracy by the interviewer.

### Data management and analysis

We cleaned and analyzed quantitative survey data using Stata software (Version 17.0, Stata-Corp, College Station, TX). We summarized the data in tables as frequencies and percentages. We summarized the Likert scale responses as bar charts using Microsoft Excel [17]. We mapped the distribution of MUST residency alumni across the region/countries of current work in QGIS (version 3.16.3) using GPS coordinates.

The interviews were analysed independently by 2 researchers using thematic analysis approach [18]. Each analyst read through each transcript several times highlighting and labelling blocks of text with related underlying meaning (codes). The identified codes were then subjected to constant comparison [19] before being merged into categories of codes with related meaning. The themes connecting the codes within each category were then identified. The 2 researchers met to discuss these themes, although there were minor differences in the labelling of themes, there was no significant differences in the themes and are reported descriptively.

### Quality control and assurance

Access to data was restricted to only the principal investigators (LT, JK, JN) who had the security to the Google drive folder, where the data was sent during data collection.

## Results

### Demographics

In total, 303 e-mails were sent out. Twenty six (8.5%) of those mails were undelivered, which left us with 277 usable e-mails, from which we received 95 completed questionnaires. This yielded a response rate of 34.3%. Of the 95 respondents who participated in the tracer survey, majority were Obstetricians and Gynecologists (38%, n = 36), followed by General surgeons (19%, n = 18), pediatricians (9%, n = 9) and Ear Nose and Throat specialists (ENT) (6%, n = 6) (Fig 1); majority were males (80%, n = 76), aged between 30–40 years (66%, n = 69), and had completed their training in the years after 2018, and had privately sponsored their post-graduate studies (36%, n = 34), others had scholarship support from fellowships, government and NGOs (Table 1).

### Geospatial distribution of post-graduate alumni of the clinical residency program of MUST

There was uneven distribution of clinical residency alumni of MUST across Uganda, with sparse distribution in Kenya and Rwanda. There was imbalanced distribution of the different specialties across the region with Obstetricians/Gynecologists (Obsgyn) and General surgeons more dispersed across the region compared to the sparse distribution of other specialties (Fig 2).

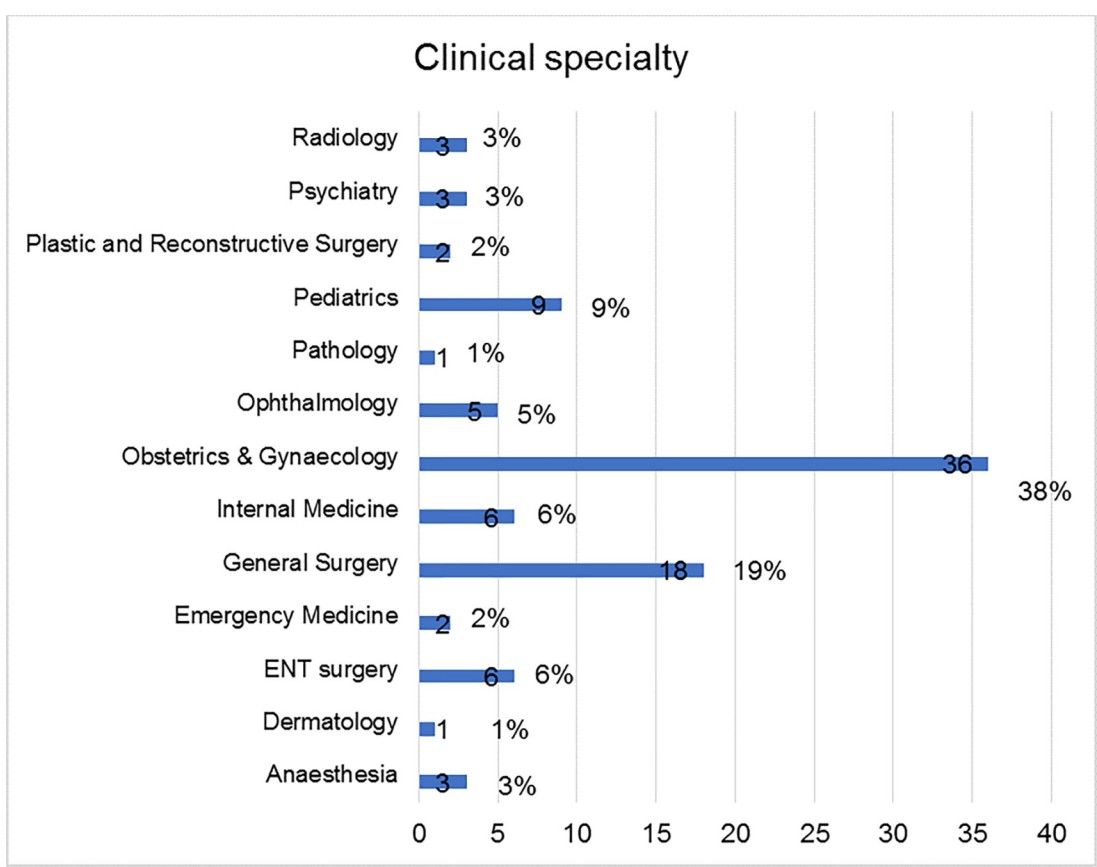

**Fig 1. Distribution of clinical specialties of the survey respondents.**

### Work place locations of post graduate alumni of the MUST clinical residency program

Majority of the clinical residency alumni are distributed in the south-western, central and northern regions of Uganda, and at the different public and private Hospitals across the East African region especially in the Urban and peri-Urban settings (Fig 3).

### Usefulness of residency training to one's career

The respondents generally reported that the residency training was to a large extent useful to their careers—in preparing them for their current work tasks (48%), and for tasks in other spheres of life (45%). Majority of the respondents reported that the current employment and work was to a large (40%) and very large (36%) extent appropriate to their level of training; and the career at the time of graduation had been realized to a large and very large extent in 37% and 33% of the respondents, respectively (Fig 4).

### Employment status of post-graduate alumni of MUST

**Obtaining of first employment.** The respondents reported the following as important or very important for being employed; personality, grades at the university, reputation of the university, reputation of the university and previous work experience (Fig 5).

**Current job employment.** All the respondents to the survey were currently employed and most were earning UGX 5–10 million (USD 1400–2800) per month (46%). Majority reported

**Table 1. Background characteristics of the respondents.**

| Characteristics | | Frequency (%) N = 95 |
|---|---|---|
| Age (years) | | |
| | 20–30 | 7 (7%) |
| | 31–40 | 66 (69%) |
| | 41–50 | 21 (22%) |
| | 51–60 | 1 (1%) |
| Gender | | |
| | Female | 19 (20%) |
| | Male | 76 (80%) |
| Nationality | | |
| | Burundi | 2 (2%) |
| | DRC | 5 (5%) |
| | Ghana | 1 (1%) |
| | Kenya | 10 (11%) |
| | Rwanda | 3 (3%) |
| | Somalia | 2 (2%) |
| | Swaziland/ Eswatini | 2 (2%) |
| | Tanzania | 2 (2%) |
| | Uganda | 68 (72%) |
| Current marital status | | |
| | Married | 67 (71%) |
| | Single | 18 (19%) |
| | Cohabiting | 8 (8%) |
| | Divorced/Separated | 2 (2%) |
| Year of residency completion | | |
| | 2006 | 2 (2%) |
| | 2007 | 1 (1%) |
| | 2008 | 1 (1%) |
| | 2009 | 1 (1%) |
| | 2010 | 1 (1%) |
| | 2014 | 3 (3%) |
| | 2015 | 6 (6%) |
| | 2016 | 2 (2%) |
| | 2017 | 10 (11%) |
| | 2018 | 16 (17%) |
| | 2019 | 18 (19%) |
| | 2020 | 17 (18%) |
| | 2021 | 17 (18%) |
| Highest qualification | | |
| | Master in Medicine (MMed) | 83 (87%) |
| | Fellowship | 9 (9%) |
| | PhD | 3 (3%) |

having obtained their first employment <6 months after graduation (76%). Majority had spent <1 year in their current employment (31%), followed by those who had spent >5 years (29%). Also majority had spent <1 year in their current position (36%), followed by those who had spent between 1–3 years (35%). Majority of the participants were currently employed in Uganda (67%), were involved in clinical practice/care (82%) and are public service employees (51%) (Table 2).

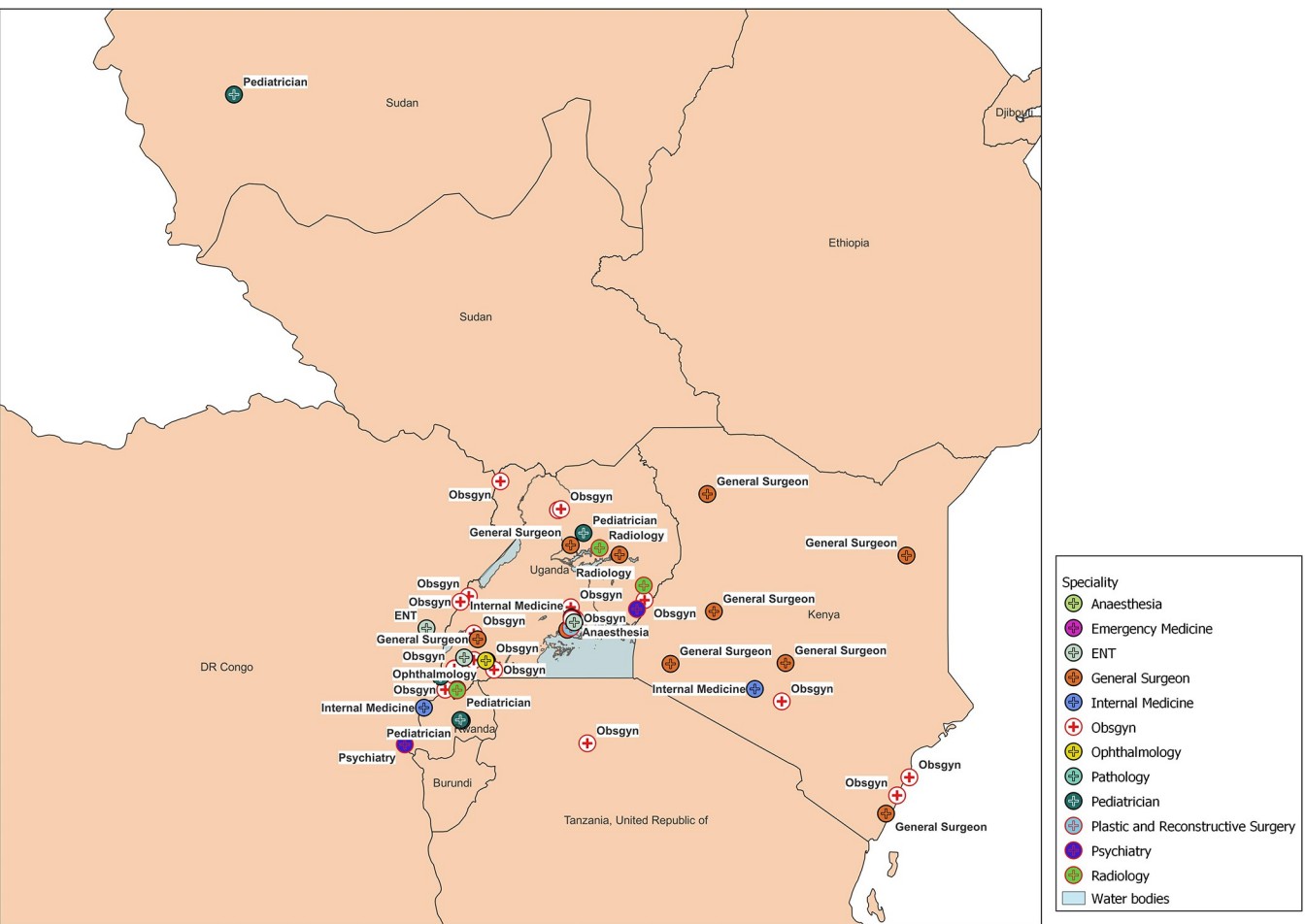

**Fig 2. Spatial distribution of clinical residency alumni from MUST and the different specialties across the East African region.** Link to map base layer: https://www.naturalearthdata.com.

## Work orientation, job satisfaction and professional success of graduates

Overall majority of the respondents were either satisfied (43%) or very satisfied (29%) with their professional situation. They were very satisfied with; possibilities to use qualifications acquired during their studies (53%), the opportunity to benefit society (42%) and the content of work/ professional tasks (37%). Majority were also satisfied with; the opportunity of pursuing continuous learning (52%), the chance of realizing their own ideas (53%), and income (47%). However were not as satisfied about the equipment at work places, position and promotion prospects and income (Fig 6). Overall, there was no significant difference in satisfaction among those in public versus private, a bigger proportion were satisfied (37% [public] versus 49% [private]) or very satisfied (30% [public] versus 29% [private sector]), (p<0.460).

## In-depth interviews

Of the participating participants, 55 (58%) agreed to be contacted to be interviewed, 12 participants were randomly selected by program cohort year to allow for a variety of perspective from the participants as the program has evolved over time. We conducted 12 semi-structured in-depth interviews with several themes regarding the almuni's post residency work

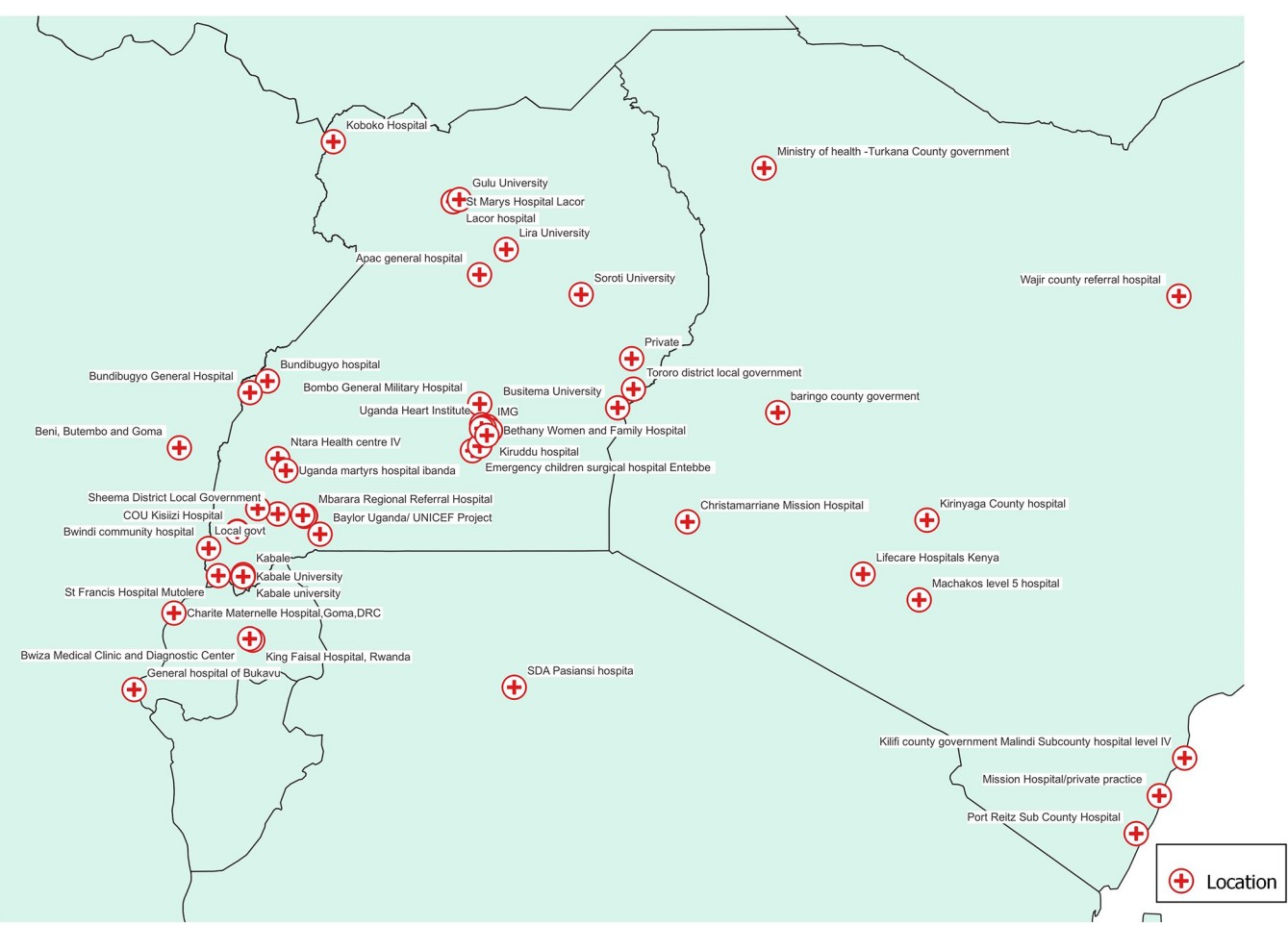

**Fig 3. Work place locations of post graduate alumni of the MUST clinical residency program across the East African region.** Link to map base layer: https://www.naturalearthdata.com.

experiences, their ability to put the skills obtained to use and impact communities they serve, and the challenges encountered during their work.

## Post residency work experiences

Processes of finding jobs included self-led job searches, and employer led identification of employees.

". . .just soon after finishing at MUST, I applied for the job and then came to the dean's office at MUST and he recommended me so I was short listed after applying and I sat for the interview and I was given the job."[IDI02]

". . .it was not a situation where it was advertised and then I had to compete, they just picked me and said that they think I can help them and so they invited me." [IDI03]

Participants reported changing job due to search for job security and for strategic positioning.

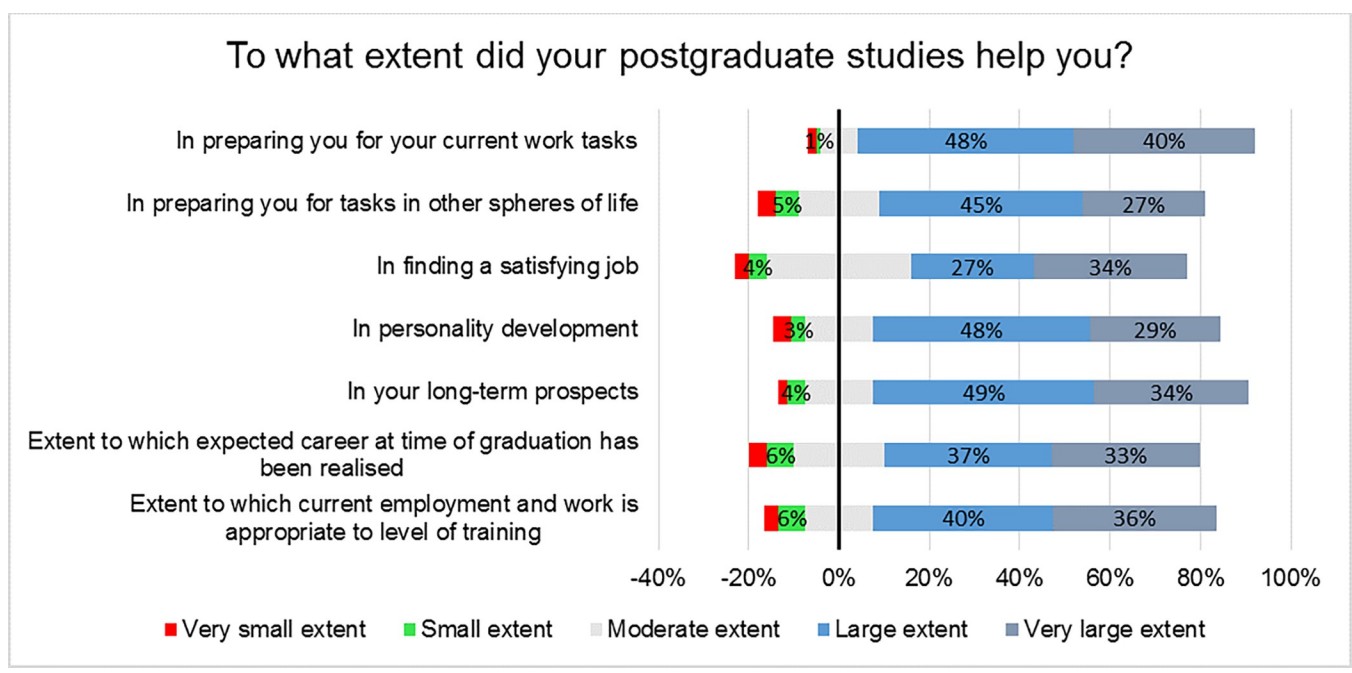

**Fig 4. Usefulness of residency training to one's career.**

"...the reason was more of getting a permanent job because KIU was a bit temporary and contract based, so I needed something more secure."[IDI06]

"...though they were paying me much better but having an opportunity to come to Mbarara under the government was even a bigger opportunity because at that time they were paying me less but I was looking at the opportunities of mentorship ad being able to come back to school with ease."[IDI01]

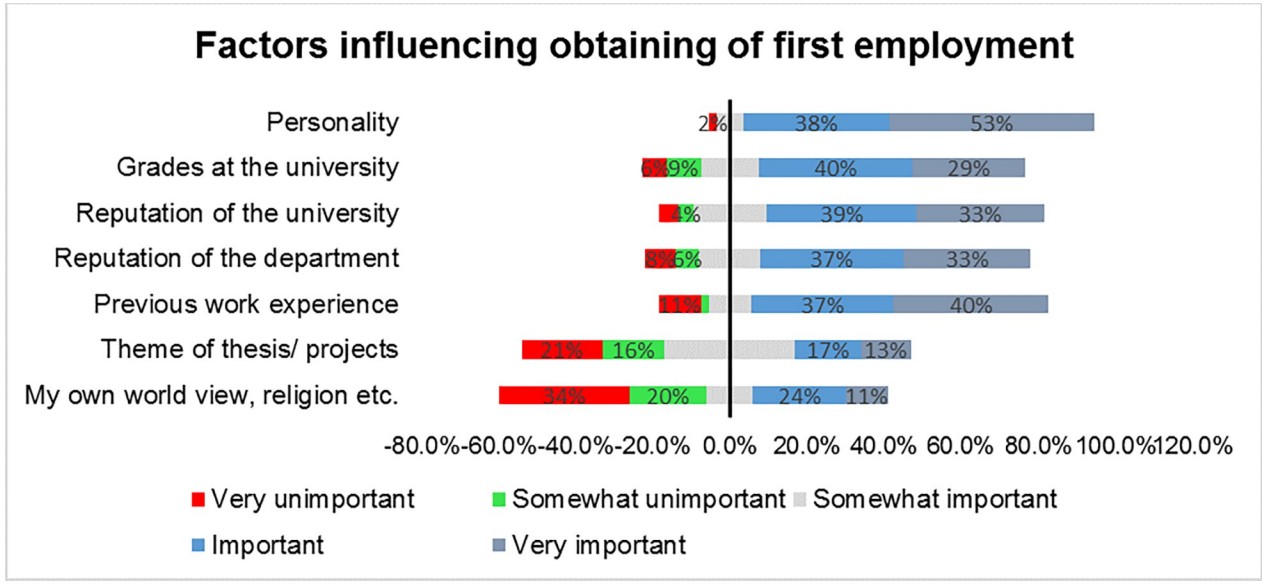

**Fig 5. Factors influencing obtaining of first employment.**

**Table 2. Current job employment.**

| Characteristics | | N = 95 |
|---|---|---|
| Time between graduation and first employment | | |
| | <6 months | 72 (76%) |
| | 6 months—1 year | 14 (15%) |
| | >1 year | 9 (9%) |
| Start of job seeking | | |
| | After graduation | 23 (24%) |
| | Around the time of graduation | 10 (11%) |
| | Before graduation | 34 (36%) |
| | I did not seek employment | 28 (29%) |
| Current employment status | | |
| | Employed | 85 (90%) |
| | Advanced academic study | 8 (8%) |
| | Self employed | 2 (2%) |
| Current area of work assignment* | | |
| | Clinical care/practice | 78 (82%) |
| | Research and development | 13 (14%) |
| | Training and teaching | 34 (36%) |
| Current employer | | |
| | Public | 48 (51%) |
| | Private | 33 (35%) |
| | Non-governmental organisation | 14 (15%) |
| Nature of current employment | | |
| | Full time | 76 (80%) |
| | Part- time | 6 (6%) |
| | Both | 13 (14%) |
| Type of contract for the current employment | | |
| | Permanent | 66 (69%) |
| | Temporary | 29 (31%) |
| Location of current job | | |
| | Uganda | 64 (67%) |
| | Kenya | 11 (12%) |
| | Rwanda | 3 (3%) |
| | Democratic Republic of Congo | 3 (3%) |
| | Somalia | 2 (2%) |
| | Swaziland/ Eswatini | 2 (2%) |
| | Tanzania | 1 (1%) |
| | Burundi | 1 (1%) |
| | Canada | 1 (1%) |
| | Ghana | 1 (1%) |
| | Botswana | 1 (1%) |
| | Madagascar | 1 (1%) |
| | Sudan | 1 (1%) |
| Duration of current employment | | |
| | Less than 1 year | 29 (31%) |
| | 1–3 years | 26 (27%) |
| | 3–5 years | 12 (13%) |
| | More than 5 years | 28 (29%) |

(*Continued*)

**Table 2.** (Continued)

| Characteristics | | N = 95 |
|---|---|---|
| Duration of work in the current position | | |
| | Less than 1 year | 34 (36%) |
| | 1–3 years | 33 (35%) |
| | 3–5 years | 14 (15%) |
| | More than 5 years | 14 (15%) |
| Change of employment/employer since your graduation | | |
| | No | 71 (75%) |
| | Yes | 24 (25%) |
| Number of times changed employment | | |
| | Once | 14 (58%) |
| | Thrice | 4 (17%) |
| | Twice | 5 (21%) |
| | More than 4 times | 1 (4%) |
| Approximate monthly gross income in UGX millions (USD) | | |
| | <UGX 5 million (<USD 1400) | 35 (37%) |
| | UGX 5–10 million (USD 1400–2800) | 44 (46%) |
| | > UGX 15 million (>USD 2800) | 16 (17%) |

*Participants involved in more than one work assignment.

## Ability to put the skills obtained to use and impact the communities served

The alumni reported relevance of their training and their ability to impact those they serve in terms of both teaching, research, management and clinical care.

> "*I think I am putting these skills to use because I am doing clinical work and I also obtained some research skills that I am using to teach students skills on how to do research work*". [IDI04]

> "*It [Training] is actually very relevant, Kabale has never had an ENT surgeon for a long time so I should say that I was the first and for the time I have been there the number of referrals from Kabale to Mbarara referral hospital greatly reduced, so I am happy that I am able to offer help not only to the academia but also to the community. So that is how relevant my training has been*". [IDI02]

> "*I would say that virtually everything that I learnt, I am trying to give it out. I am actually using all my skills because I learnt research and internal medicine and at the moment I am using all the skills I learnt, you know there are some people who get office jobs and don't use their skills but personally, I am using my clinical skills as well as other knowledge to teach*". [IDI05]

> "*My discipline is contributing to improving the lives of mothers and children and in a way helping to maintain a healthy community that will contribute to the economic development of the country*". [IDI07]

> "*I put the skills almost at a daily basis, every single day I am using the skills I learnt during my post graduate training, both clinical practice and management, I have used the skills I learnt during my post graduate*". [IDI03]

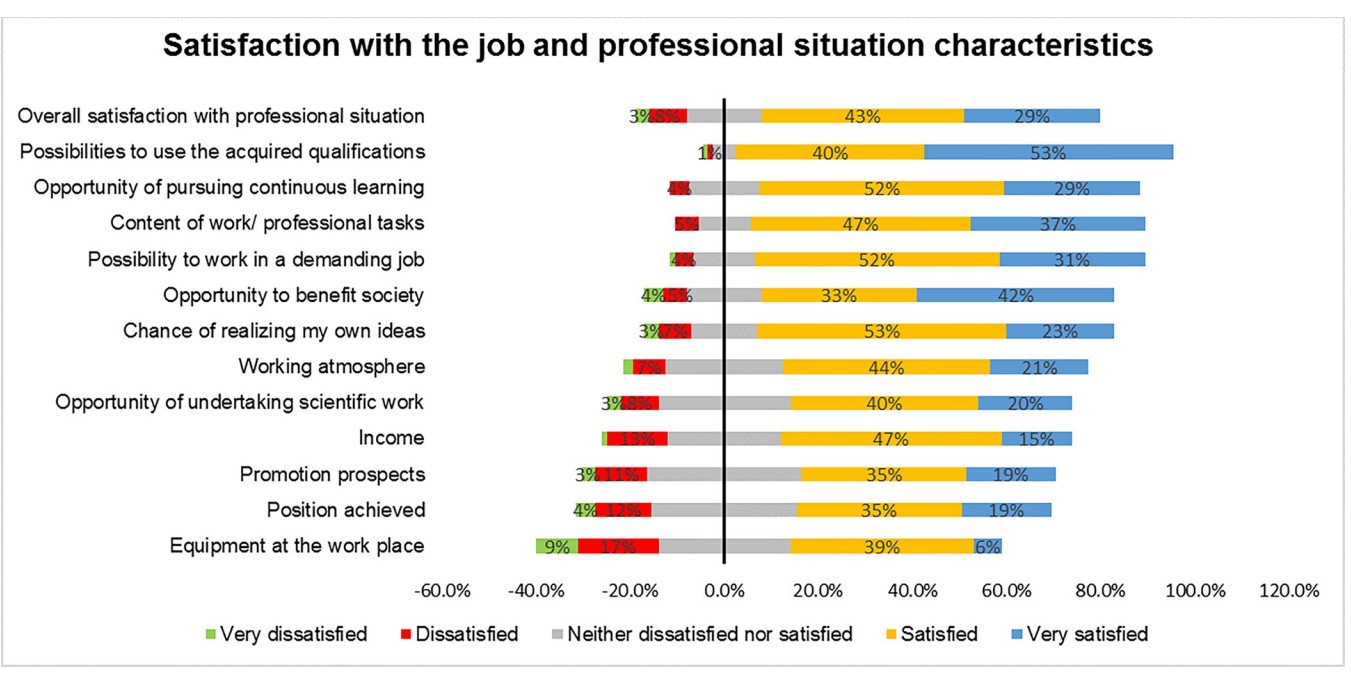

**Fig 6. Job and professional satisfaction of graduates.**

## Challenges encountered by the post-residency alumni at the places of employment

Several challenges have been encountered such as underpayment, lack of enough resources at work, delayed job promotions and challenging career development.

## Significant under payment

The alumni reported dissatisfaction with their payment compared to the work output, for example one participant reported, *"...we are not fully satisfied with the salary. It is small compared to the amount of work that we do and the training that we go through to get there..."* [IDI06]

Another participant thought, they were carrying out 2 tasks/jobs, that they perceived as separate i.e., "teaching" and "clinical care", yet they received pay for a single task; *"I am teaching, I am still working as a [general] doctor [and] as a specialist so it is likes I am doing two jobs and I am only being paid for one job."*[IDI05]

## Opportunities for career advancements at the current places of employment

The alumni reported varying opinions regarding opportunities for career advancement at their different places of employment. Some reported challenging career development while others reported favorable career growth environments. The decisions to advance were however influenced by social needs and the places of acquisition of the career advancements.

*"Currently at my institution [of work] I haven't seen much opportunity in terms of career advancement; I think the only thing I have seen is expanding my research and innovation funds which can work in the direction of career advancement, generally there is minimal opportunities."* [IDI01]

*"The opportunities are there, there are PHD scholarships and Masters Scholarships for the general staff. I think the opportunities are there".* [IDI03]

*"The opportunities are limited but the connection to the outside environment is available".* [IDI11]

*"..the opportunities are there only that the challenge is that at this time of life when you have already grown with a family, it becomes hard to progress because of many reasons because you may find that you could attend certain fellowships that need you to leave the country or to relocate which becomes difficult when you have a young family."* [IDI01]

### Need for promotions of specialists with the skills attained

The alumni expressed dissatisfaction with the promotion processes both in the academic and clinical spheres of practice, as stated below;

*"Considering the number of years I have spent in the university, the number of students I have taught and the supervisions I have done I am not happy because I have not been promoted while my fellow colleagues in other departments have been promoted, I think I should be at least at a level of senior lecturer."* [IDI05]

*"I have stayed long on the same level for the last nine years so the promotion has been less."* [IDI05]

*"I think the government can do better [regarding] promotions, because I am looking at some-one who has officially completed and was given a study leave to go and study and then he comes with papers that show he has graduated but they fail to promote people even when there is room. So there are many people functioning at lower positions, so I think the government can do better".* [IDI09]

### Inability to use the acquired skills

The alumni decried their inability to use the obtained skills, due to the unavailable facilities—equipment and medicines, which may be associated with failure to achieve the maximal productivity potential. For example one of the alumni reported, *"There are some things that you can never do in a government setting because you don't have the facilities where to do them, like accessing certain drugs and some equipment. So you end up imagining something and then that is all you can do for that but otherwise we have the patients and the cases but we are limited in that way".* [IDI01]

*"I have just been limited by equipment so there are some procedures that I can't carry out from this side due to lack of equipment but generally speaking I can say that I am putting these skills that I learnt in my daily activities, apart from those limitations".* [IDI05]

*". . .you know when you are working for the government, you find that you are short of resources which can enable us to utilize our skills. That is what is challenging."* [IDI08]

## Discussion

We explored the distribution and contribution to access to specialty care and described the integration of the clinical residency alumni of MUST into the job market, their current job satisfaction and the challenges experienced.

There was uneven distribution of the specialties among the respondents of the survey—Obstetrics and Gynecology, General surgery, Pediatrics and Internal medicine accounting for majority of the respondents to the survey. This is in keeping with the spatial distribution of the health workforce of the different medical specialties, with an imbalance between the established clinical specialties (like Obstetrics and Gynecology, General surgery, Pediatrics and Internal medicine) and the non-established specialties (Dermatology, Pathology, Emergency medicine, Anesthesia, Psychiatry)—which are sparsely distributed across the region, this may in the long-run impact health care delivery. The uneven distribution of the specialties is influenced by specialty career preferences of medical students and their eventual choice of career training as evidenced by prior studies [20–23]. There is need to avert the uneven distribution of specialties in order to ensure adequate health care delivery by attracting residents to specialties with inadequate numbers of health workforce so as to ensure a holistic improvement of healthcare.

We found that the alumni seamlessly integrated into the job market after their training—over 76% of the respondents were employed in the first 6 months of graduation and did not find a lot of difficulty in getting employed. The overall satisfaction with the professional situation was good (72%)—with majority citing usefulness of the acquired skills, benefit to society, job security and opportunity to pursue further education to be associated with their current professional situation. As much as the opportunities of employment for specialist are available, the alumni reported experiencing significant underpayment, limited resource availability for their optimum working and delayed job promotions. These are also generally the leading causes of dissatisfaction among the different human resources for health, especially in the low resource settings and therefore contribute to recurrence of industrial actions in these settings [24–26]. Strengthening health financing in terms of timely promotions, commensurate remunerations and provision of adequate resources/equipment, would improve the job satisfaction of the graduates especially in the cited areas of dissatisfaction [24–26].

There is uneven geographical distribution of clinical residency alumni of MUST in Uganda—with concentration in the south-western, central and northern regions of Uganda, and at the different private and government hospitals across the East African region. The areas of distribution mostly represent the hospitals and the training institutions in the urban centers which are usually well equipped with competitive salaries for the respective human resources. Urban areas are additionally more attractive to health care professionals, as they provide the social, cultural and professional benefits—like educational opportunities and access to amenities for their families [27,28]. This leaves a gap in the distribution of specialty services at remote rural areas including district-level hospitals, this will further improve rural access to specialist services and decrease referrals to the main tertiary and teaching hospitals. The inequalities in the training and distribution of specialist health work force has previously been reported across East and Southern Africa [1,11,12,29,30]. The difficulties in retention of health workers in the rural and remote settings, with preference for urban and wealthy areas affects both developed and is more pronounced in the developing countries and poses significant challenges to equitable health care delivery [11,12,27]. As much as there was evident inequalities in the urban versus rural distribution of the participants, over 2/3 of the specialists were employed in Uganda, this speaks to the context of specialist being trained and retained to provide clinical care, education and leadership as has been evidenced in other low resource settings, this consequently contributes to the reduction in the global health care workforce crisis [1].

The majority of participants are involved in clinical care and/or training and teaching as their major area of activity, this significantly directly and indirectly benefits the population served in the sense of provision of quality health care and in the training of additional health care workers to supplement on the already constrained health work force in our low resource

setting. During the qualitative interviews, the participants further stressed the great relevance of the skills obtained during the residency training and their ability to impact the communities they serve in the areas of clinical care, teaching, research, and management. The long term commitment to residency training provides an opportunity to strengthen specialist capacity building at the different levels ranging from the national, regional and district level in the different areas of expertise boasted by the alumni in clinical care, research, education and leadership. These positively impact on the quality of health care of the populations served [2,31].

## Limitations

The study sample was small, this is however dependent on the survey response rate. Our survey response rate was low, at 34.3%, compared to surveys conducted among doctors in general (averaging at 53%) [32], but comparable to specialist response rates to web-based surveys (35.0%)—citing the lack of time to respond to the surveys [33]. However low, our survey response rate would be considered appropriate to provide confident estimates [34,35]. Our response rate may also be explained by the fact that our survey was concurrently shared on Whatsapp platforms of the residency alumni. We additionally enriched the survey findings with qualitative in-depth interviews. The survey was voluntary and anonymous, this may be prone to sampling bias—respondents with a smooth transition to employment and in a good professional situation may be more willing to take the survey than those who had difficult transition to employment and/or poor professional situation. Our findings are from residency alumni of only a single university and may not represent the contribution to the distribution and pool of specialists from other universities. We however believe some of the experiences may be similar to those of alumni from other public universities in Uganda and the East African Community which share similar characteristics as MUST and the places of employment.

## Conclusions and recommendations

This study highlights the importance of the residency training in improving the professional situation of the alumni, through providing an opportunity for career and professional growth and improvement of the quality and safety of health care services in the communities served by the graduates. It also provides insights in the need to support training and retention of specialists in specialties with inadequate numbers of health workforce and the remote rural areas including district-level hospitals, this will reduce the imbalance in the distribution of the specialists by improving rural access to specialist services and decrease referrals to the main tertiary and hospitals—consequently improving health care delivery.

## Supporting information

**S1 Checklist. Inclusivity in global research.**
(DOCX)

**S1 Data. Dataset for impact of MUST residency training on increasing specialty workforce.**
(DTA)

## Acknowledgments

The authors acknowledge the office of the academic registrar for providing the email contacts of the residency alumni. We also acknowledge Ms. Gloria Ninsiima and Mr. Gabriel Nuwagaba for participating in data collection.

## Author Contributions

**Conceptualization:** Leevan Tibaijuka, Joseph Ngonzi.

**Data curation:** Leevan Tibaijuka.

**Formal analysis:** Leevan Tibaijuka, Jonathan Kajjimu.

**Funding acquisition:** Leevan Tibaijuka, Musa Kayondo.

**Investigation:** Leevan Tibaijuka, Jonathan Kajjimu, Asiphas Owaraganise, Musa Kayondo.

**Methodology:** Leevan Tibaijuka, Asiphas Owaraganise, Adeline Adwoa Boatin, Joseph Ngonzi.

**Project administration:** Leevan Tibaijuka, Jonathan Kajjimu, Joseph Ngonzi.

**Resources:** Nixon Kamukama.

**Supervision:** Leevan Tibaijuka, Jonathan Kajjimu, Lorna Atimango.

**Visualization:** Jonathan Kajjimu.

**Writing – original draft:** Leevan Tibaijuka, Jonathan Kajjimu, Joseph Ngonzi.

**Writing – review & editing:** Leevan Tibaijuka, Lorna Atimango, Asiphas Owaraganise, Adeline Adwoa Boatin, Musa Kayondo, Nixon Kamukama, Joseph Ngonzi.

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
