## [Decision Letter · Decision Letter 0]

18 Jan 2024

PGPH-D-23-02119

Impact of the Mbarara University of Science and Technology residency training on increasing access to specialty care workforce

Dear Dr. Tibaijuka,

Thank you for submitting your manuscript to PLOS Global Public Health. After careful consideration, we feel that it has merit but does not fully meet PLOS Global Public Health’s publication criteria as it currently stands. Therefore, we invite you to submit a revised version of the manuscript that addresses the points raised during the review process.

We look forward to receiving your revised manuscript.

Kind regards,

Abraham D. Flaxman, Ph.D.

Academic Editor

Journal Requirements:

2. Please include a complete copy of PLOS’ questionnaire on inclusivity in global research in your revised manuscript. Our policy for research in this area aims to improve transparency in the reporting of research performed outside of researchers’ own country or community. The policy applies to researchers who have travelled to a different country to conduct research, research with Indigenous populations or their lands, and research on cultural artefacts. The questionnaire can also be requested at the journal’s discretion for any other submissions, even if these conditions are not met.  Please find more information on the policy and a link to download a blank copy of the questionnaire here: https://journals.plos.org/globalpublichealth/s/best-practices-in-research-reporting. Please upload a completed version of your questionnaire as Supporting Information when you resubmit your manuscript.

3. Please provide separate figure files in .tif or .eps format only and remove any figures embedded in your manuscript file. Please also ensure all files are under our size limit of 10MB.

4. We do not publish any copyright or trademark symbols that usually accompany proprietary names, eg  ©, ®, ™  (e.g. next to drug or reagent names). Please remove all instances of trademark/copyright symbols throughout the text, including ® on page 19.

5. Some material included in your submission may be copyrighted. According to PLOS’s copyright policy, authors who use figures or other material (e.g., graphics, clipart, maps) from another author or copyright holder must demonstrate or obtain permission to publish this material under the Creative Commons Attribution 4.0 International (CC BY 4.0) License used by PLOS journals. Please closely review the details of PLOS’s copyright requirements here: PLOS Licenses and Copyright. If you need to request permissions from a copyright holder, you may use PLOS's Copyright Content Permission form.

Potential Copyright Issues:

Figs 2 & 3: please (a) provide a direct link to the base layer of the map (i.e., the country or region border shape) and ensure this is also included in the figure legend; and (b) provide a link to the terms of use / license information for the base layer image or shapefile. We cannot publish proprietary or copyrighted maps (e.g. Google Maps, Mapquest) and the terms of use for your map base layer must be compatible with our CC-BY 4.0 license. 

"

Additional Editor Comments (if provided):

Please make changes as suggested by the reviewer to emphasize that this is a convenience sample.

Reviewers' comments:

Reviewer's Responses to Questions

**Comments to the Author**

1. Does this manuscript meet PLOS Global Public Health’s publication criteria? Is the manuscript technically sound, and do the data support the conclusions? The manuscript must describe methodologically and ethically rigorous research with conclusions that are appropriately drawn based on the data presented.

Reviewer #1: Yes

2. Has the statistical analysis been performed appropriately and rigorously?

Reviewer #1: Yes

3. Have the authors made all data underlying the findings in their manuscript fully available (please refer to the Data Availability Statement at the start of the manuscript PDF file)?

Reviewer #1: Yes

4. Is the manuscript presented in an intelligible fashion and written in standard English?

Reviewer #1: Yes

5. Review Comments to the Author

Reviewer #1: General

This is a clear, well-written manuscript. It’s main contribution is in describing a convenience sample of providers that were trained in a residency program. I think the authors could emphasize they have a convenience sample a bit more clearly and earlier in the manuscript, because this could explain the positive affect of the responses. Regardless of this limitation, I think it makes some nice points about HRH satisfaction, geographic distribution of providers following graduation and the reasons for satisfaction or not.

Specific

Abstract: you should include the number of qualitative interviews here too. I would describe your sample as a convenience sample in the abstract as well.

How do your response rates compare to other provider surveys?

It is likely that you have (self)-selection bias in both the online survey participation as well as the in-depth interview participation and I would postulate it biases the results toward those more satisfied with the program. How might you address this selection bias or do so in future studies? For example, phone follow-up would be a way to increase response rates. The low response rates will be important to mention this as a limitation of the study.

Is there anything about the program that emphasizes OB/GYN? Or is this a feature of the job market? Providers’ interests?

Are there differences in job satisfaction between providers in the public vs. private sectors?

Figure 6: you should complete this label: “Possibilities to use qualifications acquired during my…”

Quotations: There are a few typos you should address.

6. PLOS authors have the option to publish the peer review history of their article (what does this mean?). If published, this will include your full peer review and any attached files.

**Do you want your identity to be public for this peer review?** For information about this choice, including consent withdrawal, please see our Privacy Policy.

Reviewer #1: No

---

## [Decision Letter · Decision Letter 1]

23 Feb 2024

Impact of the Mbarara University of Science and Technology residency training on increasing access to specialty care workforce

PGPH-D-23-02119R1

Dear Dr. Tibaijuka,

We are pleased to inform you that your manuscript 'Impact of the Mbarara University of Science and Technology residency training on increasing access to specialty care workforce' has been provisionally accepted for publication in PLOS Global Public Health.

Best regards,

Abraham D. Flaxman, Ph.D.

Academic Editor

Reviewer Comments (if any, and for reference):

Reviewer's Responses to Questions

**Comments to the Author**

1. If the authors have adequately addressed your comments raised in a previous round of review and you feel that this manuscript is now acceptable for publication, you may indicate that here to bypass the “Comments to the Author” section, enter your conflict of interest statement in the “Confidential to Editor” section, and submit your "Accept" recommendation.

Reviewer #1: All comments have been addressed

2. Does this manuscript meet PLOS Global Public Health’s publication criteria? Is the manuscript technically sound, and do the data support the conclusions? The manuscript must describe methodologically and ethically rigorous research with conclusions that are appropriately drawn based on the data presented.

Reviewer #1: Yes

3. Has the statistical analysis been performed appropriately and rigorously?

Reviewer #1: Yes

4. Have the authors made all data underlying the findings in their manuscript fully available (please refer to the Data Availability Statement at the start of the manuscript PDF file)?

Reviewer #1: Yes

5. Is the manuscript presented in an intelligible fashion and written in standard English?

Reviewer #1: Yes

6. Review Comments to the Author

Reviewer #1: Thank you for making the suggested revisions.

7. PLOS authors have the option to publish the peer review history of their article (what does this mean?). If published, this will include your full peer review and any attached files.

**Do you want your identity to be public for this peer review?** For information about this choice, including consent withdrawal, please see our Privacy Policy.

Reviewer #1: No
